# Understanding the Dynamics of the COVID-19 Pandemic: A Real-Time Analysis of Switzerland’s First Wave

**DOI:** 10.3390/ijerph17238825

**Published:** 2020-11-27

**Authors:** Marina Giachino, Camille Beatrice G. Valera, Sabina Rodriguez Velásquez, Muriel Anna Dohrendorf-Wyss, Liudmila Rozanova, Antoine Flahault

**Affiliations:** Institute of Global Health, University of Geneva, 1202 Geneva, Switzerland; liudmila.rozanova@unige.ch (L.R.); antoine.flahault@unige.ch (A.F.)

**Keywords:** COVID-19, SARS-CoV-2, pandemic, Switzerland, emerging infectious diseases

## Abstract

Since the novel coronavirus outbreak of SARS-CoV-2 from the first cases whereof were reported in Wuhan, China, in December 2019, our globalized world has changed enormously. On the 11th of March 2020, the World Health Organization (WHO) declared COVID-19 a pandemic, and nations around the world have taken drastic measures to reduce transmission of the disease. The situation is similar in Switzerland, a small high-income country in Central Europe, where the first COVID-19 case was registered on the 25th of February 2020. Through literature review as well as correspondence with public health professionals and experts in mathematical modeling, this case study focuses on the outbreak’s impact on Switzerland and on the measures this country has implemented thus far. Along with the rapid spread of the virus, the political organization, economy, healthcare system, and characteristics of the country greatly influence the approach taken in facing the crisis. Switzerland appears to be structurally well-prepared, but, according to mathematical modeling predictions, in order to avoid total collapse of healthcare facilities, the measures taken by the Swiss Government need to reduce the virus transmission chain by at least 70%. Fortunately, updated models on April 22nd show evidence that the non-pharmaceutical measures invoked have decreased transmission by an estimated 89%, proving effective management by the federal government and allowing for progressive deconfinement measures.

## 1. Introduction

Outbreaks of emerging infectious diseases (EIDs) are circumstances in which countries should be prepared to manage, yet world leaders continue to forewarn a health crisis that will reveal the vulnerabilities of the global framework. In December 2019, accumulating reports of individuals diagnosed with pneumonia of unknown etiology in Wuhan, Hubei Province, China, marked the beginning of an EID. Named severe acute respiratory syndrome 2 (SARS-CoV-2), the novel pathogen shares multiple characteristics with viruses within the Coronaviridae family previously encountered, such as SARS-CoV in 2002 and MERS-CoV in 2012. While clinical presentations are similar, the exponential growth of coronavirus disease 2019 (COVID-19) cases overwhelmed healthcare facilities and aggressively spread beyond Chinese borders [1]. As the crisis progressed, the World Health Organization (WHO) officially elevated the situation from a Public Health Emergency of International Concern (PHEIC) to a pandemic on the 11th of March 2020.

Societal frameworks of affected countries have been pushed to their limits as they struggle to adapt and suffer consequences beyond health impacts. Switzerland in particular faces unique challenges, since it is a landlocked country with neighboring regions such as Italy [2], which has experienced alarmingly high rates of COVID-19. In addition, Switzerland hosts many international headquarters and employs a multitude of cross-border workers, increasing the risks of population exposure and transmission of disease.

Following Switzerland’s first confirmed case of COVID-19 in the canton of Ticino on the 25th of February 2020 [3], multiple cases related to the Italy clusters were quickly discovered. Thereafter, progressively strict measures were imposed by the Federal Council in efforts to contain the spread and reduce strain on national resources. No official lockdown was implemented in contrast to other countries, but the Federal Council issued declarations banning social gatherings and urging individuals to remain home as of 16th March 2020. One month later, the Federal Council announced that confinement measures would slowly be lifted using a three-phase strategy beginning on April 27th due to improvement of epidemiological trends.

The objective of this case study is to provide a holistic and factual overview of the response to the COVID-19 pandemic in Switzerland.

## 2. Methodology

We conducted a “real-time analysis” as the pandemic was still an ongoing event as of June 2020. Although the collection of data and results of this pandemic is continuously evolving, the information presented in this case study is relevant, as it depicts the state of the pandemic in Switzerland at the time of the study’s publication.

We utilized and gathered sources such as official websites from the Swiss Confederation, published articles, and correspondence with public health professionals and experts between March and April 2020. The collected data were from accredited and reliable websites and articles. For the report, we researched and utilized country-specific information and data related to the COVID-19 pandemic, making the statistics and numbers portrayed only applicable to Switzerland. Additionally, the information on neighboring countries was obtained to provide an international comparison of the responses to the pandemic.

This case study defines the beginning of the COVID-19 situation in Switzerland with the first case of SARS-CoV-2 in Switzerland (February 25th of 2020); therefore, the sources used for this report directly relating to the COVID-19 pandemic in Switzerland are dated as early as 25th February. Since the study considers the information ranging from the arrival of COVID-19 in Switzerland up to the date of its publication, the last accessed date of the sources is 3rd June 2020. Consequently, this case study is considered a “real-time” analysis of Switzerland’s pandemic as the information from the event is taken in the middle of the pandemic. Therefore, much of the data and possible outcomes continue to evolve.

## 3. Findings

### 3.1. Case Presentation

#### 3.1.1. Characteristics of Switzerland

Switzerland is a landlocked country in Central Europe surrounded by regions of France, Germany, Austria, Liechtenstein, and Italy, making it more susceptible to the import of any infectious disease within its borders. Its territory comprises a total surface area of 41,285 km^2^ characterized by three main geographic regions: the Alps, which cover nearly 60% of the country, the Swiss Plateau (30%), and the Jura (10%). Switzerland’s geographic location [4] and its complex topography also greatly influence the climate profile which in turn impacts the pattern of certain climate-sensitive infectious diseases. For instance, the Alps represent a striking climatic border between the northern and southern regions. Winters in the North are mild and humid, while at higher altitudes, cold temperatures with snow are the norm. In contrast, the southern side of the Alps is strongly influenced by the Mediterranean Sea, where winters are mostly mild with humid summers ranging from warm to hot [5].

A geographically small country, Switzerland is composed of 26 cantons with four national languages (German, French, Italian, and Rhaeto-Romanic). To ensure coexistence between the various language, regional, and cultural differences, Switzerland features an original government system. A federalist state since 1848, the nation’s political system is strongly influenced by federalism and direct democracy. The Federal Constitution sets out the tasks that the Confederation and the cantons must fulfill, emphasizing the freedom of choice and self-determination of the population [6]. The total Swiss population reached 8.5 million in 2018, of which 25% are foreigners, and a total of 85% of its inhabitants live in urban areas [7]. The life expectancy of the Swiss population is 85.4 years for women and 81.7 years for men, one of the highest worldwide (2018). In 2019, 18.7% of the Swiss population was 65 years old or older [8]. This is important demographic information when it comes to the epidemiological data on COVID-19, since vulnerable populations are elderly and the most affected are men over 65 years of age.

Switzerland is also an interesting business location for companies and multinational corporations. Small and medium-sized enterprises (SMEs) constitute most Swiss firms and are of great importance to the economy, notably in the export sector. Foreign trade is another critical component to the Swiss economy. The country’s main trading partner is the European Union (EU), with 52% exports and 70% imports [9]. Additionally, the Swiss labor market employs cross-border commuters, a majority of whom work within the Lake Geneva region and the canton of Ticino. Among the 328,000 cross-border commuters in 2019, a total of 55% came from France, 23%—from Italy, and 18%—from Germany [10]. Switzerland’s impeccable workforce is considerably responsible for its strong economy. The service sector alone is responsible for approximately 74% of the gross domestic product (GDP) with a 25% contribution from produced goods. In 2018, the GDP per capita increased by an estimated 2.8% year-to-year and stood at 81,000 Swiss francs (million CHF) [11]. The Gini coefficient (distribution of income) reached 32.7% in 2017 with a slight decrease compared to 2016 (33%) [12]. In addition, 44% of the 25–64-year-olds in Switzerland hold a tertiary degree, in comparison with an average of 39% in the OECD countries. Among the younger generation, the share of tertiary education is higher (2018) [13].

Overall, Switzerland’s unique makeup and characteristics are components which country officials must acknowledge when determining what measures to implement during the COVID-19 pandemic.

#### 3.1.2. The Swiss Healthcare System

The healthcare system in Switzerland is regulated by the state and private sector. The Swiss Confederation is responsible for legislation and control of the compulsory health insurance, communicable diseases, as well as of medically assisted reproduction and organ transplantation. Supervision of healthcare, hospitals, prevention, regulation of licenses to practice health professions and top-level medicine is regulated at the federal and cantonal level. Primary basic insurance is compulsory for all persons, who purchase it from private insurance companies, and a “supplementary health insurance” covering other areas (dentistry, complementary medicine, etc.) [14] is optional.

Total health expenditures of 82,774 (million CHF) represents 12.4% of Switzerland’s GDP, with health expenditures per capita amounting to 816 CHF per month (reported in 2017). A total of 281 hospitals and 38,000 hospital beds are reported in Switzerland, and 1,468,000 hospitalization cases were recorded in 2018 [15].

There are several health ranking systems available. We decided to look at three ranking systems: WHO ranked Switzerland number 20 in the overall efficiency of the 191 WHO member states. The ranking system from 2000, although outdated, contains various factors (five health system goals): health, health inequality, responsiveness–level, responsiveness–distribution, and fair-financing [16]. Furthermore, Switzerland reached rank 16 in the evaluation by Bloomberg in 2014 (out of 51 countries). It was measured in terms of cost as percentage of the GDP (11.38%) and cost per capita (8980 per person in US $) [17]. In addition, Switzerland was evaluated in a recent ranking system and reached the first rank of 11 countries: the health system performance was graded in three different domains: general performance, clinical outcome, and sustainability indicators. This ranking system is shaped to test the outcome performance and the stability of the different healthcare systems. Overall, the general performance and clinical outcome were rated excellent, and in the context of healthcare system equity and sustainability, the Swiss healthcare system is considered above average. In other terms, according to this ranking system, the Swiss health system is of high quality and can sustain economic instability (Prof. Antoine Flahault, personal communication). Factors of the healthcare system and preparedness might be an advantage in the fight against the COVID-19 outbreak in Switzerland.

#### 3.1.3. Epidemiological Situation of the Country Regarding COVID-19

In order to obtain a holistic overview of the current epidemiological situation occurring in Switzerland, it is crucial to consider epidemiological data at the global, national, and cantonal level, since emerging infectious diseases such as COVID-19 are unaffected by borders. Although the SARS-CoV-2 outbreak was initially recorded to have started late December in Wuhan, China, the outbreak did not reach the Swiss ground until February 25th when the first viral case identified by Swiss authorities was classified as an imported case. The infected patient reported traveling to the city of Milan, Italy, where a high prevalence rate of COVID-19 was observed. Soon after the first reported case, the number of cases increased throughout Switzerland, making the prevalence of COVID-19 cases 34.2 per 10,000 inhabitants as of May 3rd [18]. As of May 3rd, the country experienced a total of 29,905 cases of COVID-19, of which 3643 cases were active and 26,262 closed. Across the active cases, 3502 were reported to have mild symptoms and 141 were serious, requiring intensive care. A positive recovery rate of 81.9% and fatality rate of 5.9% have been reported in Switzerland [19]. Each Swiss canton is in charge of reporting the confirmed cases, making the reporting rate and accuracy of the data vary within cantons. It is important to keep in mind that the reported cases differ from the actual number of cases as patients infected with SARS-Cov-2 might develop mild symptoms or be asymptomatic and might not seek medical attention or testing. This makes it hard to portray an accurate number of cases, and therefore obtain appropriate epidemiological data needed for the correct estimation of inputs for modeling purposes. By taking a cantonal look at the impact of COVID-19 in Switzerland, we can observe that in terms of total reported cases, the canton of Vaud experienced the highest total cases of COVID-19 (5284 cases); however, a more accurate depiction of the situation is to use the prevalence of cases per 10,000 people in each canton. When using this parameter, we notice that the canton of Geneva experienced the highest prevalence of cases (99.1) compared to the cantons of Ticino (91.6) and Vaud (66.1) on 3rd May [18]. Looking at the epidemiological data from 11 June, a major decrease in the active cases of COVID-19 has been recorded since the peak of the pandemic. Only a total of 375 cases are considered active as of 11 June. This demonstrates that the pandemic is at its last stages, as a much smaller percentage of the population is appearing to be actively sick with SARS-CoV-2.

Epidemiological curves based on the data from the European CDC regarding the daily number of new reported cases of COVID-19 were created for Switzerland and its four main neighboring countries (Italy, France, Germany, and Austria). The graphs are shown below.

Based on Figure 1, differences and similarities between the daily numbers of COVID-19 cases per 100,000 inhabitants reported in Switzerland and its neighboring countries can be noted. In terms of differences, the case rate in Switzerland is relatively higher than in its neighboring countries due to the high number of cases per 100,000 seen in late March and the beginning of April. Similarly, all the five countries started reporting their first cases of COVID-19 at the end of February. Additionally, the highest peak of reported cases during the first wave similarly ranged from the end of March to the beginning of April in all the five countries.

### 3.2. Management and Outcomes

#### 3.2.1. Non-Pharmaceutical Intervention Measures Undertaken by Health Authorities

To tackle the novel coronavirus disease (COVID-19), the Swiss Federal Council has taken drastic action and implemented an ordinance with the objective of ordering measures that will reduce the risk of transmission and combat the virus. Its purpose is to prevent or contain the spread of COVID-19 in Switzerland, reduce the frequency of transmission (interrupt transmission chains and prevent or contain local outbreak), protect particularly endangered persons, ensure Switzerland’s capacity to cope with the epidemic, and in particular, maintain the conditions for an adequate supply of medical care (in particular, intensive care beds) and therapeutic products to the population [20].

The main federal measures undertaken by the health authorities in order to fight against COVID-19 are listed in the timeline below (Figure 2).

The Swiss strategy emphasizes social distancing measures, especially the mixing of younger generations with older ones, to avoid a complete and official lockdown of the population. The Federal Council focused on reducing the number of contacts between individuals in order to slow the spread of the virus by banning all private and public gatherings as well as closing schools and restricting passage at borders. The Swiss Government calls for citizens’ good behavior and common sense to understand the situation and follow proper hygiene measures by keeping a minimum of two meters distance, washing hands thoroughly, avoiding physical contact, coughing and sneezing into a paper tissue or the crook of the arm. This particularly applies to individuals belonging to high-risk populations, such as people over 65 years of age and those with chronic or pre-existing conditions who were instructed to restrict themselves at home during the confinement phase [21]. This was also an unusual situation for the army, since it is the first time after World War II (WWII) that Switzerland called up thousands of army reservists to help eliminate the current COVID-19 crisis. The additional workforce and supplies are allocated throughout the areas of health, logistics, and security [22]. The situation was exceptionally difficult for the healthcare system, which needed to be somehow rearranged. Healthcare provisions from the Federal Council granted cantons authority to oblige private hospitals and clinics to accept patients. All clinics, medical and dental practices were also instructed not to perform non-urgent operations and treatments. Since the COVID-19 outbreak, hospitals have been preparing for a crisis situation under immense pressure. Following the directive from the Federal Council, there is close cooperation between private and public hospitals, healthcare personnel have been retrained to cope with the outbreak, and medical practices have been adapted. Hospital staff who had already been overworked prior to the outbreak, faced the regulations of suspended rest periods or working time restrictions (previously, 60 h). In order to combat coronavirus, the Federal Council temporarily suspended the Labor Code for the “hospital wards experiencing a massive increase in work as a result of COVID-19 diseases” [20].

#### 3.2.2. Observed and Expected Impacts on the Country Economy

Healthcare systems are not the only sector affected and overwhelmed by the pandemic. Since the beginning of the SARS-CoV-2 outbreak, disruption in the Chinese industry significantly impacted the global market. China is one of Switzerland’s key trading partners (5% of Swiss imports, 8% of exports in 2017) [23], and the fall in Chinese production had a notable influence on the nation’s economy. Various sectors—manufacturing in particular—were left exposed considering the turnover reduction and fragmented supply chains. The Swiss economy was further interrupted as the virus encroached its borders, causing many small and medium-sized enterprises to reduce or cease operations altogether. Federal mandates over the course of the viral spread also had crippling effects on tourism and service industries due to major event cancellations, ski resort closures, and flight disruptions as other countries initiated travel bans. For instance, large-scale events such as the Geneva International Motor Show were expected to stimulate local economies based on the substantial number of attendees, but now represent drastic loss in cantonal revenues.

Despite the current state of the global market and an estimated GDP loss of 30 billion CHF, the Minister of Economic Affairs, Guy Parmelin, was initially optimistic and announced that the Swiss market still functioned at 80% capacity on March 20th [24]. As of April 8th, however, Parmelin disclosed that latest estimates show a 25% loss of production with 30% of Switzerland’s workforce in short-time work, and unemployment has risen to almost 3% [25]. This is in stark contrast to the Swiss unemployment rate of 2.3% in the beginning of 2019, which was reportedly at the lowest rate in 20 years. [26] To curtail downward economic spiral in the early stages of the pandemic, the Federal Council authorized a 10 billion stimulus package and subsequently added 32 billion CHF days later to be allocated for company bank loans, protection of jobs, support for self-employed individuals, and honoring wages amongst other measures. Additional funds for economic relief were approved as the crisis persisted for a total of 65 billion CHF [27]. Compensation rights have also been extended to temporary workers, apprentices, and parents whose work lives were interrupted as a consequence of school closures. Moreover, the Federal Council approved deactivation of the counter-cyclical capital buffer which strengthens the stimulus package measures and essentially increases the lending authority of banks to fulfill growing needs of individual households and businesses [28]. Nevertheless, the Federal Government’s Expert Group stated that recession is inevitable, as the economic disturbance of COVID-19 has initiated a recurrent pattern of downsizing, terminations, and decreased consumption. More ominously, the Federal Expert Group warns of “the biggest slump in economic activity since 1945” with a projected GDP decrease of 6.7% and significant rise in unemployment rates, the yearly average of 3.9% in 2020 and 4.1% in 2021 [29].

With the improvement of Switzerland’s epidemiological situation, further economic impacts are likely to be minimized in light of the three-phase deconfinement framework issued by the Federal Government. Provided that COVID-19 cases in the country continue to stabilize within the upcoming months and stringent measures are not reinstated, economists roughly predict that Swiss markets will begin a fragile recovery in 2021 as production slowly recommences. However, recovery of the Swiss economy is also highly dependent on the resilience of foreign markets. Considering the state of the international economic environment as well as the probable decrease in private and public expenditures, experts forecast that the GDP during Switzerland’s 2021 recovery will increase by 5.2%, not even reaching the GDP observed in 2019 [29].

On the other hand, officials state that recovery may not be as dismally predicted, since current forecasts remain uncertain due to limited data. For example, the assumption of severely restrained consumer behavior may be false, whereas the return of normal private expenditures will allow for a quicker recovery of economic losses. Furthermore, foreign trading partners may not be as vulnerable as perceived and indeed have the capacities necessary to strongly resume business activities, incrementally leading to the restoration of the national economic cycle.

#### 3.2.3. Social or Political Disruption and (Social) Media Coverage

Another consequence of the COVID-19 pandemic is the significant social and political disruption it presents to countries. Switzerland as a federal state is even more exposed to political disruption, especially after the pronouncement of the “extraordinary situation,” on the 16th of March 2020. Briefly, it means that the Federal Council may alone, without consulting the cantons, order measures for all or part of Switzerland. By doing so, the democratic process in this country is hindered by the coronavirus and the Federal Council takes predominant control of the crisis. In addition, COVID-19 generates a strong social disruption. The prevention measure of social distancing is undoubtedly required during this crisis, along with this significant change in social norms through the abolishment of handshakes, hugs, etc. Thus, according to the lengths of the social isolation, there may be mental health repercussions in the near future as physical contact is often needed [30].

During the COVID-19 crisis, the media coverage is described as remarkably important and the need for quality information has never been greater. According to C. Bergstrom, Biology professor at the University of Washington in Seattle, this virus sprouted “the first social media pandemic” [31] because of the heavy reliance of the population on social media for information. Using mass media for the purpose of disclosing information, avoiding fake news or a social media panic [32] is a means that is often adopted. For example, the Geneva University Hospital (HUG) is employing Instagram to post explanatory stories on the novel coronavirus. Likewise, the Swiss retail markets posted reassuring messages that counteract a food shortage. By the same token, the audience figures following the newscast seem to rise constantly [33], as the Swiss population is eager to know about the daily measures taken by the Federal Council. Even the Federal Counselor in charge of health, Alain Berset, launched a call on Instagram to eradicate the coronavirus and asked for help from prominent individuals, who responded immediately with the hashtag #protectyourselfandothers.

#### 3.2.4. Mathematical Modeling Predictions

Tools such as mathematical models are needed to guide government policies as they are one of the best ways to understand, visualize, and quantify any pandemic. When modeling predictions for Switzerland and the world, the SEIR model, susceptible (S), exposed (E), infected (I), and recovered or removed (R), creates the best predictions for the current COVID-19 pandemic. This model allows countries to adequately plan for the pandemic through the use of predictions on the speed of the spread of SARS-CoV-2 in the population, as well as assess the effectiveness of control strategies. Additionally, other compartments such as hospitalization and critical care (ICU) can be added to the model in order to create a useful dynamic transmission model, allowing authorities to make adjusted decisions. However, in order to obtain a better picture of the current epidemic in Switzerland, the use of reported deaths suggests a better predictive model than using the number of infected cases. This is because the number of cases is subjective to the testing rate of each canton and mild symptoms or asymptomatic cases are underreported by the authorities. While the death rate of COVID-19 provides a better model, the total number of cases cannot be inferred from the death count, as no reliable estimate of fatality rate exists. This adds an additional layer of uncertainty and inaccuracy in the epidemiological data used for any mathematical models created. According to the work of Dr C. Althaus on March 25th at the Institute of Social and Preventive Medicine at the University of Bern (ISPM) [34], the basic reproduction number (R_0_) of COVID-19 was estimated to be at 2.99 for Switzerland before the social distancing measures were implemented by the country. For the epidemic to slow down and eventually disappear, the value of R_0_ would require a reduction of at least 67% in order to be considered below the critical threshold of 1 and allow for the return of societal normalcy. Mathematical predictions obtained from Dr Althaus’ work on March 25th seen in Figure 3, provided the different epidemiological trajectories obtained when the transmission of the virus is reduced by different percentages such as if the transmission of the virus is not reduced (100% or 1), 25% reduced (0.75), 50% reduced (0.5), 60% reduced (0.4), 70% reduced (0.3), 80% reduced (0.2), 90% reduced (0.1), and completely reduced (0% or 0) after the implementation of social distancing measures applied by the government. As demonstrated in Figure 3, the interruption of transmission plays an important role in reducing the number of deaths and infections caused by SARS-CoV-2.

Switzerland reportedly has a total of approximately 1200 ICU beds available for the entire Swiss population [35]. This means that in order to avoid the straining of healthcare facilities by early April 2020 and during the peak of the pandemic, the current social distancing measures adopted by the government had to reduce virus transmission by 70%. Fortunately, when looking at the updated mathematical model of Dr Althaus from April 7th seen in Figure 4, it can be observed that the social distancing measures implemented on March 17th were able to decrease the viral transmission of SARS-CoV-2 amongst the Swiss population by 78%, allowing for healthcare facilities to remain functional and efficient [36]. These results put emphasis on the important impact non-pharmaceutical measures such as social distancing have on the reduction of transmission and therefore the number of deaths, hospitalizations, and patients requiring ICU. Additionally, due to the decrease in the viral transmission of SARS-CoV-2 down to 89% as of April 22nd, the new relaxation of the social distancing measures announced by the Federal Council had started to take place as of April 27th throughout Switzerland. Nevertheless, mathematical modelling and monitoring of the situation in Switzerland remain crucial, as it is necessary to understand the impact deconfinement measures will have on the viral transmission and the possible number of new cases.

## 4. Discussion

The COVID-19 pandemic has put the world’s political, financial, social, and health sectors to test. This has clearly been the case of Switzerland, a small country confined by some of Europe’s biggest and most influential countries. The reality of this pandemic began for Switzerland when the first confirmed case of coronavirus was reported in Ticino on the 25th of February 2020. Although each canton tried to contain the rapid spread of the virus by implementing cantonal measures, the exponential spread in the country required immediate response from the Federal Government to mitigate the transmission of the virus and thereby delay the epidemic peak of COVID-19. Since vaccines protecting the susceptible population are not expected to be available in the near future [37], the main issues COVID-19 raised in Switzerland are related to the effectiveness of governmental measures, the possible stress on available healthcare facilities, and the foreseen drastic economic challenges. Additionally, if herd immunity is not achieved for SARS-CoV-2 through vaccination or previous exposure, a second worldwide wave of infection must be feared. Based on previous epidemiological factors, various potential scenarios are possible for the progression of the SARS-CoV-2 pandemic. Following previous influenza and coronavirus outbreaks, three possible future scenarios for the COVID-19 pandemic are estimated. The first scenario consists of a series of smaller repetitive waves that occur through the summer and consequently over a 1–2-year-long period. These small repetitive waves will gradually diminish sometime in 2021. This first scenario requires periodic reinstitution and relaxation of non-pharmaceutical measures through the next 1 to 2 years of small waves [38]. The second scenario deals with a larger wave in the fall or winter of 2020, accompanied by smaller subsequent waves in 2021. This scenario is based on the pattern seen during the 1918–1919 Spanish flu pandemic and requires full reinstitution of mitigation measures to drive down the transmission of infection and prevent healthcare facilities and systems from becoming overwhelmed [38]. Finally, the last scenario explores the idea of a slow burn. This signifies that ongoing transmission and case occurrence will become the norm. This specific model provides for geographic variations and may be influenced by the various degrees of mitigation of the virus [38].

In order to properly address the needs of the country in terms of prevention and treatment for COVID-19, accurate data on the current positive cases detected in Switzerland are crucial. However, factors such as selective testing and an accelerated number of positive cases have made it difficult for the country to provide accurate figures. In the case of Switzerland, the testing rate became a controversial topic after the Federal Office of Public Health had made the statement that individuals with symptoms who had not been part of a vulnerable population and had not required medical care would no longer be systematically tested [39]. Some researchers, as well as the WHO’s recommendation on increasing the testing rate, encouraged Switzerland to urgently change its testing strategy to a more liberal approach in order to detect more infected people and provide higher economic and social benefits to the country. Despite all the recommendations and criticism received on the current testing methods, Switzerland was able to administer more than 32,670 tests per million inhabitants [40], classifying the country as one of the top performers of tests globally. This is reportedly due to the increase of testing at the cantonal level. Some cantons, such as Berne, have decided to provide higher availability of tests by including a drive-through system that allows qualifying residents to be tested from the comfort of their car [41]. High testing rates provide the necessary care and data needed for researchers and public health workers to make adjusted decisions on measures in order to flatten the epidemiological curve. Nevertheless, a current debate on strengthening surveillance methods and case tracking through possible use of contact tracing is being discussed amongst the FOPH. For Daniel Koch, head of the Communicable Diseases Division of the FOPH, the aim of using anonymous telephone data is not to establish a surveillance system as done in other countries, but to retrospectively understand how people move and how long they are in contact with others with the aim to evaluate whether the measures taken by the Federal Council are effective [42].

An increase in the testing rate also plays a major role in the accuracy of mathematical models. These can prove to be a useful tool in the prediction of cases and estimating the impact implementation measures could have on decreasing infections and deaths. Models also prove to be effective in estimating the number of hospitalized patients and patients in the ICU, allowing countries such as Switzerland to prepare available ICU beds. Since the possible shortage of ICU beds and collapse of hospitals are some of the main issues faced by Switzerland, the government emphasized strengthening healthcare facilities. Switzerland has reported to have around 11 ICU beds per 100,000 population, meaning that approximately 1000 to 1200 ICU beds are available for the entire population of Switzerland [35], a number that, for the moment, proved to be sufficient for the number of complicated COVID-19 cases. This is where recommendations such as social distancing, confinement, and improved hygienic behaviors play a crucial role in flattening the curve, as demonstrated by Dr Althaus’ work, allowing for relieving some stress on the healthcare system and facilities. As the Swiss population has demonstrated throughout the past few weeks, all these measures seem to be understood and accepted by most of the population. Yet, the positive impact of confinement measures on COVID-19 mortality must also be analyzed in the context of adverse economic consequences and all-cause mortality rates. For instance, the worsening prognosis of patients who forgo routine or emergency health services due to economic hardship or fear of contracting COVID-19 may possibly be attributed to increased mortality from other causes, such as complications of chronic diseases. Redirection of hospital resources and decreased hospitalization rate of non-COVD-19 patients may also contribute to higher all-cause mortality rates. Nevertheless, Switzerland is surrounded by European countries following a different strategy in implementing protection measures against COVID-19. Italy issued a nationwide lockdown on March 9th, ordering its residents to stay at home, Austria required compulsory face masks in supermarkets, Spain declared a state of emergency on March 14th, issuing a general confinement, and the French government announced a strict lockdown on March 17th [43]. In France, before the announcement of the nation’s lockdown, an online survey had been undertaken where 70% out of 1000 respondents estimated that a total quarantine would be appropriate given the risk and only 30% thought it would be an overreaction and an excessive measure [44], which proves the population’s agreement on the measures taken. Considering these facts, it is legitimate to wonder if adapting stricter confinement measures would have been more efficient for Switzerland. Despite the residents showing preference for the implementation of stronger measures to expedite reduction in R_0_ and decrease the severity of new infected cases, most contacted experts consider the measures taken by the Swiss government appropriate and effective, proving to be plausible as the deconfinement phase was adopted prior to the anticipated timeframe. The Federal Council regularly reiterates the importance of protecting the vulnerable population (persons over 65 years of age and people with chronic preexisting conditions) and emphasizes the relevance of Swiss solidarity. Social media have also shown to have a strong impact on the national and international solidarity, which is especially needed in these times of crisis where all social gatherings, restaurants, and educational institutions have experienced limitations and restrictions to their operating capacities and hours. Within a matter of weeks, the nation’s social and working environment changed, overturning the system and greatly impacting the economy.

Measures taken for travel restrictions and border lockdowns have strongly influenced the daily life of the Swiss and cross-border workforce. Government-imposed health preventive measures have directly impacted the labor market by halting productivity, resulting in the liquidation of companies and increase in unemployment. In order to cope with the unraveling economic consequences, Swiss Government officials approved 65 billion CHF to be allocated in part for self-employed individuals as well as for the protection of jobs. While officials were initially optimistic regarding the state of the Swiss economy, prolonged confinement measures and strain on the national resources as billions of CHF had been injected into the economy inevitably resulted in a national deficit. As of June 2nd, reports from the State Secretariat for Economic Affairs (SECO) stated that the GDP already decreased by approximately 2.6% in the first quarter due to restricted economic activity resulting from the confinement measures [45]. Multinational companies would perhaps be able to successfully recover in the aftermath of COVID-19; however, it brings into question if many small and medium enterprises, which constitute the majority of Swiss firms and its productivity, as well as start-up firms will be able to survive. In fact, the number of companies who applied for seasonal unemployment were alarming in the beginning of the outbreak in March, and numbers are only expected to increase until 2021. Furthermore, the unprecedented impacts of the COVID-19 pandemic have severely disrupted international trade and the global economy as a whole. This is certainly an issue not only for the national GDP and future economic recovery, but for importing and exporting essential medical equipment for the health sector as well. Access to stocks of masks, gowns, disinfectants, and ventilators has been difficult, as COVID-19 cases surged globally, though Switzerland has been able to gracefully manage the crisis with additional supplies from the Swiss army. However, neighboring countries such as Italy and Spain lack the necessary resources; there, medical staff were treating possibly infected patients without proper protection [46]. Disruption in international trade may introduce additional negative health outcomes as other nations experiencing staggered COVID-19 waves require aid or medical equipment, but are unable to receive the assistance due to logistics or weakened capacities worldwide. As national resources are drained and health expenditures surpass the predicted budgets for 2020, a global recession is surely expected.

## 5. Conclusions

The situation of the COVID-19 pandemic is one of dynamic changes not only in epidemiological numbers, but also in the measures implemented, as no current vaccine against the coronavirus is available. Nevertheless, lessons from the Swiss governance during such times can be observed in country preparedness, implemented measures, and its compliance to the International Health Regulations (IHR 2005). Following the COVID-19 outbreak in Switzerland, the Swiss government undertook decisive measures to eradicate the coronavirus and reduce the transmission rate through personal protective hygiene, social distancing, and travel-related measures, all of which have been well-accepted by the majority of the population and proven to be effective in alleviating the stress COVID-19 has put on healthcare facilities, as shown in Figure 3. Even though neighboring countries such as Austria and Germany were the first ones to slowly relax their provisional measures, the procedures in Switzerland were maintained until April 27th when the number of cases significantly decreased allowing for an effective transition into the post peak and deconfinement stage of the pandemic. Switzerland has also managed to comply, to a great extent, with the international recommendations such as the IHR by keeping close contact with the WHO; however, compliance with the travel restrictions can still be debated.

While the current crisis brings into perspective the importance of stronger public health systems and the fragility of our existing social frameworks, it can be argued that Switzerland was able to satisfactorily handle the number of rising cases through measures such as social distancing and effectively allow healthcare facilities to function during high operational demands. Switzerland’s success in mitigating the COVID-19 situation within its borders also accentuates the importance of universal health coverage and social security. National solidarity to the Swiss Confederation recommendations versus official mandates can be attributed to the strength of the Swiss healthcare system as well as the swift actions taken to mobilize economic resources in order to protect the interests of corporations and the public. With assurance that their wellbeing and livelihoods are secured by the existing framework, individuals are more inclined to comply with the implemented measures, as they do not have to choose between their health or earning income. Proof of federal management success is also demonstrated through the establishment of deconfinement measures earlier than originally projected as cases of COVID-19 steadily decline.

Nonetheless, positive impacts of the confinement measures can also be weighed against their negative impacts. The economy and mortality related to other diseases are also affected, especially for individuals who did not seek necessary urgent care because of the implemented measures or fear of being infected, which can lead to complications, late diagnosis, or mortality. Furthermore, Switzerland could have acted earlier in terms of implementation of non-pharmaceutical interventions rather than waiting approximately three weeks after the first reported case in the country. In fact, the relevance of the global health governance is highlighted during times of uncertainty when emerging infectious diseases and novel viruses such as SARS-CoV-2 cross borders and bring disorder worldwide. As the Director-General of the WHO Dr Tedros Adhanom Ghebreyesus stated, “this is the time for facts, not fear, this is the time for science, not rumors, this is the time for solidarity, not stigma”.

## 6. Illustrations

The Federal Office of Public Health recommends simple measures to prevent the spread of the new coronavirus. In Figure 5, the first picture is the campaign to implement the protection measures, the second is the campaign of the first deconfinement phase, and the third is the campaign of the last deconfinement phase.

## Figures and Tables

**Figure 1 ijerph-17-08825-f001:**
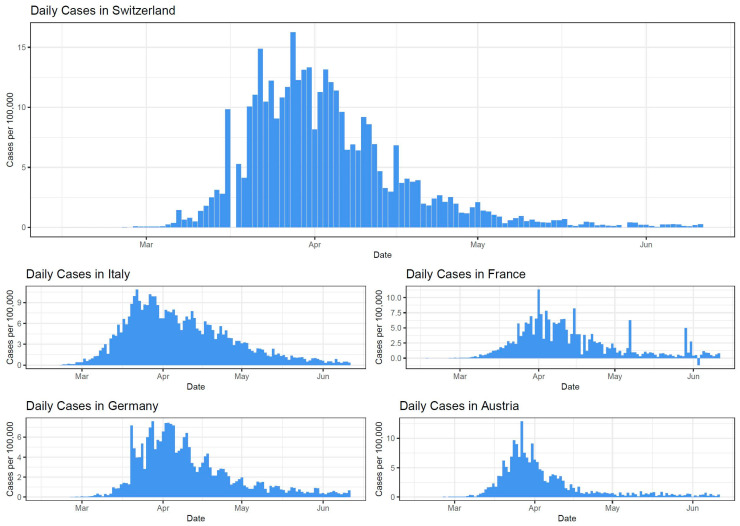
The epidemiological curves of Switzerland, Italy, France, Germany, and Austria’s daily numbers of new reported cases of COVID-19 per 100,000. The epi-curve starts on 15 January 2020, and ends on 11 June 2020. Based on the data from the European CDC.

**Figure 2 ijerph-17-08825-f002:**
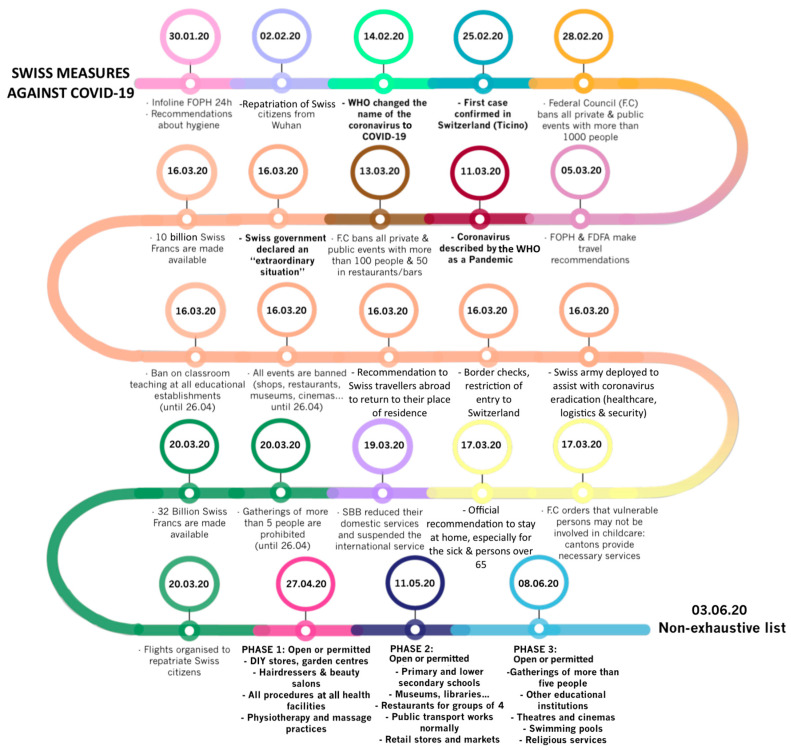
The timeline with the main dates and measures implemented in Switzerland to fight COVID-19. It is a non-exhaustive list updated on the 3rd of June 2020. Colors correspond to the dates.

**Figure 3 ijerph-17-08825-f003:**
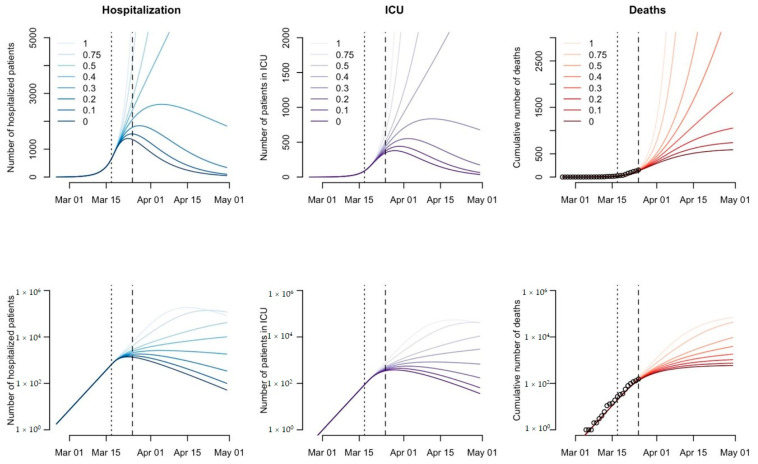
Projected numbers of hospitalizations, patients in ICU, and deaths under different scenarios for the COVID-19 epidemic in Switzerland shown on a linear scale (top panels) and a logarithmic scale (bottom panels). Vertical dotted and dashed lines indicate the time points of the social distancing measure (17 March 2020) and the last data point (25 March 2020). Source: Althaus, Christian, “Real-time modeling and projections of the COVID-19 epidemic in Switzerland.”

**Figure 4 ijerph-17-08825-f004:**
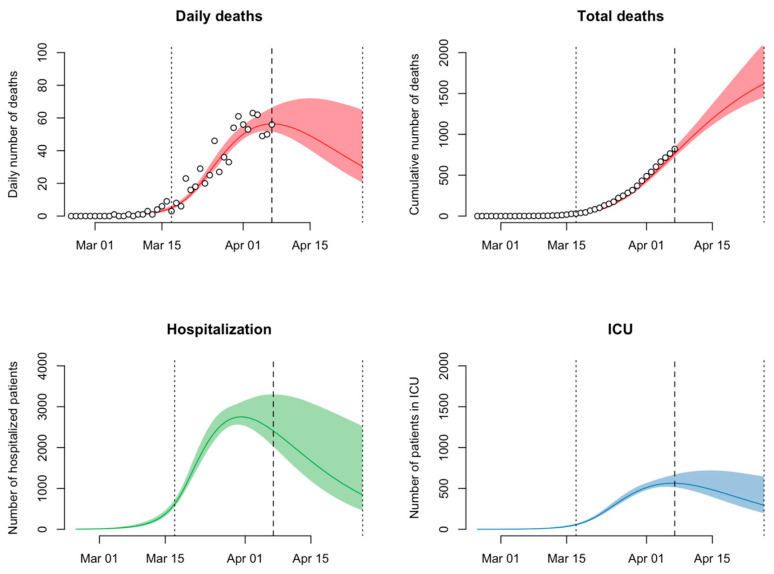
Projected numbers of daily deaths, total deaths, hospitalizations, and patients in ICU for the COVID-19 epidemic in Switzerland shown on a linear scale (top panels) and a logarithmic scale (bottom panels). Vertical dotted and dashed lines indicate the time points of the social distancing measure (17 March 2020) and the last data point (7 April 2020). Source: Althaus, Christian, “Real-time modeling and projections of the COVID-19 epidemic in Switzerland”.

**Figure 5 ijerph-17-08825-f005:**
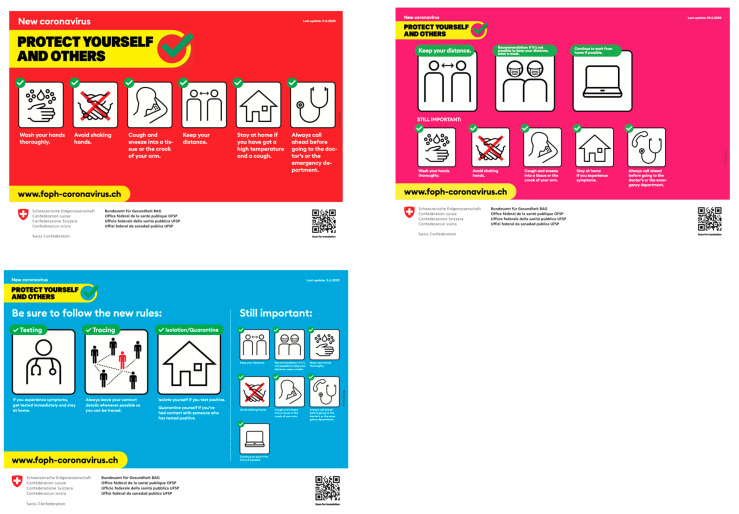
**Federal Office of Public Health** (FOPH) image campaign “Protect yourself and others;” available online [47,48,49].

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
