# Peer review of "Understanding the Dynamics of the COVID-19 Pandemic: A Real-Time Analysis of Switzerland’s First Wave"

_ijerph, 2020, doi:10.3390/ijerph17238825_

Round 1
Reviewer 1 Report
This manuscript is interesting in describing the situation of COVID-19 in Switzerland. This manuscript would benefit from significant editing - it would be suitable as a short communication - and being updated to include the current situation in Switzerland.
- Given the manuscript is under review in October/November, the authors should add additional information from June-October, or ensuring the paper reflects the current time period
- Figures 2 and 3 are from a different publication and should not be included in this manuscript. A graph of the epidemic curve in Switzerland would be more beneficial for the reader. And a comparison of the epidemic curve of Switzerland compared to neighbouring countries
- Is the Swiss people generally supportive and compliant with the directives to stay at home etc?
- Current thinking indicates that countries with economies that have been most severely impacted also have high mortality. Presumably as there is fear amongt the population. It would be useful to include this aspect into discussion about the economy.
- Given a significant second wave of infection is currently being experienced in many Northern hemisphere countries (UK, USA), what is the situation in Switzerland? Is it likely that the government enact the same measures as were enforced in March/April?
Line 42/43 – reference for overwhelmed outbreak in China
Line 48 – please include a refernce for the outbreak in Italy
Line 52 – please include the date of Switzerland’s first case
Line 69-72 – text is unclear
Line 81 – title ‘core part’ is unclear. Perhaps ‘findings’ would be more suitable?
Lines 100-105 – what proportion of the population is >65y?
Lines 06-118 – what proportion of Swiss are tertiary educated? Is this higher/equivalent to other countries?
Reviewer 2 Report
Giachino and colleagues have put together an informative manuscript that details Switzerland’s approach to the COVID-19 pandemic. They give a broad description of Switzerland’s position in the world (demographics, geography, government) that gives context to how they managed to reverse the Ro from 2.99 to <1.0 (78% reduction in viral transmission was achieved). I have a few minor suggestions.
What does the colour change in the timeline depict in Figure 1?
In figure 1 you state “expected” outcomes in phase 1, 2 and 3. Surely the actual outcomes or known now? Do they differ from the expected outcomes of the time?
Line 233 what was unemployment before covid. To get a better sense of the impact.
Line 360, it would seem we are currently going through scenario 2. Perhaps the authors could update the discussion to reflect what is currently happening with lockdown in France, Germany, the UK etc.
Round 2
Reviewer 1 Report
I thank the authors for addressing my comments.